# MolJET: Multimodal Joint Embedding Transformer for Conditional De Novo Molecular Design and Multi-property Optimization

## Abstract

Multi-property constrained optimization of molecules using generative *de novo* design models is vital for the successful application of Artificial Intelligence (AI) towards materials and drug discovery. Yet there remains a gap between the reported performance of such models in the literature and their practical utility in real world design scenarios. Furthermore, existing models are largely inaccessible to chemists without an extensive background in computer science. To address these challenges, we propose a generative foundation model, the **M**ultimodal **J**oint **E**mbedding **T**ransformer (MolJET), which performs conditional generation of desired molecular distributions based on human-interpretable chemistry prompts in a zero-shot manner. We assess MolJET on the standard benchmarks available in the *GuacaMol* and *MIMOSA* evaluation frameworks. These include structure-based sampling tasks as well as a range of multi-property optimization tasks that probe a models ability to design drug-like molecules given realistic property constraints. We demonstrate that with self-supervised pretraining, MolJET outperforms *80%* of task-optimized models while using zero-shot inferences and beats *all* baselines after minimal supervision. Moreover, the performance of MolJET on text-only conditioning tasks improves with the inclusion of property modalities during training, highlighting the importance of a multimodal approach to molecular design. MolJET is the first example of text-based *de novo* molecular design using large-scale multimodal foundation models and should serve as a building block towards further improvements to accessible AI for chemists.

## 1 Introduction

Emerging crises in climate, disease and human health threaten to permanently disrupt global stability and must be actively met with creative solutions. Many such solutions are dependent on the rapid discovery of innovative functional materials or novel drug-like molecules with optimal properties. For instance, the viability of using redox-flow batteries (RFBs) for long-term and large-scale energy storage is contingent on finding stable redox species with fast electrochemical kinetics, a feasible redox potential and high solubility (Zhang et al., 2018). Due to the immense size and complexity of chemical phase space (Polishchuk et al., 2013), the search for suitable materials is far from trivial and traditional "direct" design approaches based on iterative modifications to existing chemical structures are often far too slow (Kuhn & Beratan, 1996).

To address this issue, researchers have increasingly begun to look towards generative *de novo* design models to efficiently navigate the vast molecular phase space (Meyers et al., 2021). These models are evaluated on their ability to generate a diverse array of novel molecular structures while simultaneously biasing them towards a desired property distribution (Polykovskiy et al., 2020). Due to the ubiquity of string-based molecular representations (Weininger, 1988; Krenn et al., 2020), recent innovations in natural language modeling have been successfully applied to *de novo* molecular design. For instance, transformer architectures have achieved state-of-the-art results on property prediction tasks that require quantum-level accuracy (Ross et al., 2021) and have also been shown to increase the diversity of candidates sampled from machine-learned molecular distributions (Dollar et al., 2021).

Figure 1: **MOLJET Framework**. Prompts are (i) stochastically sampled from the available modalities in the dataset and (ii) used to condition autoregressive reconstruction of SELFIES strings. Conditions are then chosen during inference to (iii) shift the generated molecular distribution towards the desired structural or physicochemical properties.

Aside from string-based representations of molecular structures, there are other textual modalities which could provide additional context to generative models and thus improve their performance. Such modalities include IUPAC names, molecular formulas, descriptions of important chemical moieties or functional groups and natural language descriptions of chemical behavior. Yet despite the large overlap between architectures used for natural language modeling and molecular sequence modeling, there have only been a few attempts to incorporate more than a single modality within a model (Rothchild et al., 2021; Sun et al., 2021; Zeng et al., 2022) and none have included the capacity for property-driven molecular design. Massive scaling has also been primarily limited to property prediction tasks (Honda et al., 2019; Chithrananda et al., 2020) despite growing evidence of the performance benefits derived from increasing model sizes, dataset sizes and compute across all downstream tasks (Kaplan et al., 2020; Hoffmann et al., 2022).

In this work we introduce MOLJET, a large-scale multimodal joint embedding transformer for conditional molecular generation and multi-property optimization. Within this framework, molecular generation is conditioned by text-based prompts that control the structural and physicochemical characteristics of the desired molecular distributions as depicted in Figure 1. We demonstrate conditional generation on three modalities - textual descriptions of molecular structural features, physicochemical properties and 1D atomistic molecular graphs - and provide a general framework for the inclusion of additional modalities during pretraining.

To prove the efficacy of our models in realistic design scenarios, we evaluate MOLJET on a diverse set of tasks including molecular rediscovery, similarity and substructure-based sampling, isomer generation and multi-property optimization (Brown et al., 2019; Fu et al., 2021). With only self-supervised pretraining, MolJET outperforms all task-optimized baseline models on five out of the eight task categories and outperforms the baselines on all eight task categories after minimal task-specific supervised optimization. Furthermore, the prompts are designed to be easily interpretable by chemists without any prior knowledge of deep learning and thus accessible to a wider audience. We provide access to our pretrained models through an online API and hope to encourage increased participation in AI-driven *de novo* molecular design among scientific researchers in much the same way that DALL-E and GPT have inspired increased interaction with deep learning models among the general public (Brown et al., 2020; Ramesh et al., 2022).

## 2  RELATED WORK

**Multi-Property Optimization.** Several strategies for multi-property optimization of molecular structures have been explored to date. Some works propose to condition the generation of molecular structures with a learnable embedding corresponding to the values of one or more desired properties (Lim et al., 2018; Li et al., 2018; Gebauer et al., 2022). These models jointly learn the conditional

distributions during training and then allow for the selection of specific conditions during inference. Others treat optimization as a translation task, in which an improved version of the input molecule is reconstructed during training (Jin et al., 2018a; 2020). These models learn the desired molecular distribution directly, however they also require the construction of translation pairs which can be time-consuming and without careful control can introduce biases into the model or result in posterior collapse (Jin et al., 2018b). Another popular strategy for optimization is by making stepwise modifications to an existing molecular structure through an efficient sampling method like Markov Chain Monte Carlo or a reinforcement-learning driven policy network (Nigam et al., 2019; Khemchandani et al., 2020; Fu et al., 2021). A reward function determines the success of the model and guides further modifications. These models are flexible as they can modify their actions based on any reward, however they often shift the generated distribution too far from the original and can struggle to generate realistic samples (Popova et al., 2018; Brown et al., 2019).

**Foundation Models for Chemistry.** Given that the vast majority of *de novo* molecular design models operate on a single molecular representation, there are only a few examples of multimodal learning in the field of chemistry. KV-PLM and CHEMET both combine structural representations of molecules with natural language, the former by embedding SMILES strings directly into a biomedical corpus and the latter by performing cross-modal attention between embeddings of a molecular graph and a description of the molecule (Sun et al., 2021; Zeng et al., 2022). However, these models are better suited for classification tasks than generation tasks as it is challenging to build a corpus annotated with molecular structures that is large enough to train a generative model. Other examples of multimodal chemistry models include GeomGCL (Li et al., 2022) which performs contrastive learning on 2D and 3D molecular graphs for property prediction and VJTNN (Jin et al., 2018b) which combines junction tree and atomic graph representations during the encoding and decoding of the latent vector in a VAE.

## 3 Model Framework and Prompt Designing

Herein, we describe the **M**ultimodal **J**oint **E**mbedding **T**ransformer (MOLJET), a large-scale generative foundation model for conditional molecular design and multi-property optimization. The aim of MOLJET is to efficiently navigate the molecular phase space while simultaneously reaching a desired property distribution. This task is non-trivial as the molecular landscape is high dimensional and rugged making optimization within this space difficult (Stumpfe et al., 2020). We hypothesize that jointly learning across text, molecular structure and properties will enhance the model's ability to learn structure-property relationships and thus improve its performance at designing optimized molecules. We introduce the multimodal fusion with our prompt design framework in Section 3.1, and then present the model architecture and conditional sampling scheme in Sections 3.2 and 3.3, respectively.

### 3.1 Multimodal Fusion with Prompt Designing

Our goal is to learn inter-modal and cross-modal information with an expressive prompt design that can facilitate both the self-supervised pretraining and zero-shot evaluation. We propose an *early-fusion* strategy to jointly reason over the text, molecular structure, and property modalities with a shared multifaceted representation. We represent the textual description and associated physicochemical properties of a molecule in the prompt sequence $x = (s_1, s_2, .., s_n)$ of the form $(s_{text}, s_{prop}, s_{mol})$,

```
<text_type>...</text_type> <text>..</text> <property>..</property> <val>..</val> <mol>..</mol>
```

We include `<text_type>` and `<property>` tags to differentiate across molecule descriptions ($s_{text}$) and properties ($s_{prop}$). The `<text>` and `<val>` tags designate the search space on the respective data modalities. The `<mol>` tag designates the SELFIES string describing the molecular structure ($s_{mol}$). The proposed prompt design is flexible so that other textual representations of molecules or associated properties may be easily substituted. We also allow each modality to contain multiple sub-prompts. For example, we can represent multiple physicochemical properties separately as sub-prompts in $s_{prop}$. We introduce a strict ordering of the prompt sequence with the corresponding text, property and molecular structure representations to enable the model to conditionally generate molecular distributions given the other modalities.

## 3.2 MODEL ARCHITECTURE

Our objective is to pretrain a large-scale foundation model with the ability to generalize to unseen tasks without requiring any labeled data. This is specially relevant in molecular design scenarios where we need to generate new molecules that have not been previously seen (*out-of-distribution generalization*). However, it is intractable to enumerate across all possibilities due to the unbounded molecular search space. We present the unsupervised distribution estimation $p(x)$ from a set of prompts $(x_1, x_2, .., x_n)$ as the product of conditional multimodal token probabilities,

$$p(x) = \prod_{i=1}^{n} p(s_n | s_1, .., s_{n-1}) \tag{1}$$

Our model design is inspired by the recent success of applying the transformer encoder architecture on shared mulitmodal multifaceted representations (e.g., UTF-8 bytes in Perceiver-IO (Jaegle et al., 2021), vision-language decoding (Aghajanyan et al., 2022)). In this work, we investigate whether transformer architectures are capable of learning over multimodal molecular information and translating it into a rich knowledge of the relationship between a molecule's structure and its properties. We seek to analyze whether transformer architectures are suitable to distill and accumulate both inter- and cross-modal information from the molecular descriptions, and test whether the pretrained models generalize to novel contexts during *de novo* molecular design.

To this end, we adopt the autoregressive transformer decoder model architecture similar to GPT-3 (Brown et al., 2020) and apply it on conditional multimodal prompt based molecule generation tasks. We translate the general left-to-right language modeling objective to a joint modeling objective that predicts the next modality token. We minimize the joint loss defined as

$$\mathcal{L}(\theta) = \frac{1}{|D^{train}|} \sum_{x \in D^{train}} -log p_\theta(s_i | s_{\leq i}) \tag{2}$$

The model learns the conditional multimodal token distribution jointly given the in-context references to other modality tokens. We do not use modality-specific encoders in this setup since we translate all modalities into the discrete language space. It remains as a future work to explore how other modalities such as vision (continuous), graph (2D) or atomic coordinates (3D) could be used in our framework to further enrich the learned multimodal molecular representations.

## 3.3 CONDITIONAL MOLECULE GENERATION

Given the molecular structure represented as a sequence of tokens describing the atoms, their connectivity and their valence states $(m_1, ..m_n)$, the conditional multimodal prompt-based molecule generation is as follows:

$$\hat{m} \approx arg \max_{m} log\, p_\theta(m_t | s_{text}, s_{prop}, m_{<t}) \tag{3}$$

We use $q$ temperature sampling to autoregressively sample the SELFIES tokens $m_t$ conditioned on the multimodal prompt. The sampling takes the molecule textual description $s_{text}$, physicochemical properties $s_{prop}$ and `<mol>` $\in m_{<t} \subset s_{mol}$ as the initial inputs in the joint multimodal embedding space. In addition, the molecule generation is conditional to the property values in $s_{prop}$.

$$m_t = q(\cdot | s_{text}, s_{prop}, m_{<t})$$
$$s_{mol(t)} = \cup_{m_{<t} \in s_{mol(t-1)}} \{(m_{\leq t} \circ m_t^n) | m_t^n)\}_{n=1}^{N} \tag{4}$$

We sample N molecule tokens until we reach a `</mol>` tag. The sampled tokens are concatenated $\circ$ with other top scoring molecule tokens to generate the molecule structure $s_{mol(t)}$.

## 4 EXPERIMENTAL SETUP

### 4.1 IMPLEMENTATION AND TRAINING DETAILS

**Dataset Creation.**   We gathered over 100M unique molecular structures from the PubChem compound records database (Kim et al., 2019) to use for pretraining. Each structure includes a valid SMILES representation, an IUPAC[1] name, and a molecular formula. Functional groups are extracted from the full IUPAC name and SMILES are encoded as SELFIES strings. In accordance with the method outlined in GuacaMol (Brown et al., 2019), we calculate the ECFP4 fingerprints (Rogers & Hahn, 2010) for every molecule in our dataset and a holdout set of drug-like molecules used in the benchmarks. Any molecule in the training set with a tanimoto fingerprint similarity of $\geq 0.343$ to any molecule in the holdout set is removed. This ensures the model has not simply memorized solutions to the benchmark tasks during pretraining. Similarly, all isomers corresponding to the two isomer generation tasks were also removed from the training set.

Conditional prompts for each molecule are generated stochastically so the model may only see a portion of the available modalities for any given sample. This allows the user to ignore some modalities during inference while still allowing the model to jointly learn over all possible modalities. The rules for prompt sampling are outlined in Appendix B.

**Available Modalities.**   We provide three modalities on which the models are conditioned - textual molecule descriptions, properties and 1D atomistic molecular graphs. Table 1 shows the submodalities available for the text and property modality types. Each text type provides a different level of detail regarding the molecular structure and are all commonly used by chemists when describing molecules. The properties are selected to cover a wide range of chemical behavior important to drug design. Each property is calculated using the cheminformatics package RDKit (Landrum et al., 2013) aside from DRD2 which is predicted by the model published in Olivecrona et al. (2017). We use SELFIES as our 1D atomistic molecular graph to guarantee the validity of all molecules generated during inference (Krenn et al., 2020).

Table 1: Details of the multimodal inputs used in the pretraining and zero-shot evaluation.

| | | |
|---|---|---|
| **Textual Molecule Descriptions** | **IUPAC**, text that fully specifies the atomic connectivity of the entire molecule | **FuncGroups**, text that specifies only the atomic connectivity of local environments within the molecule | **MolFormula**, text that does not specify any connectivity information but does specify the overall atomic makeup of the molecule. |
| **Physicochemical properties** | **Topological polar surface area (TPSA)**, a measure of the overall surface polarity of the molecule (Prasanna & Doerksen, 2009) | **LogP/Penalized LogP (PLogP)**, a method for estimating the solubility of a molecule (Wildman & Crippen, 1999).PLogP includes penalties for molecules with low synthesizability | **BertzCT**, a topological index meant to quantify the "complexity" of a molecule (Bertz, 1981) |
| | **QED**, a quantitative measure of the "drug-likeness" of a molecule (Bickerton et al., 2012) | **Number of fluorine atoms**, **Number of aromatic rings**, **Total number of rings** | **DRD2**, the biological activity of a molecule towards the dopamine receptor $D_2$ |

**Tokenization**   We develop a custom vocabulary that consists of the tokens representing the molecule textual description $s_{text}$, physicochemical properties $s_{prop}$ and molecular structure $s_{mol}$. IUPAC and FuncGroups share a vocabulary learned from a byte-pair encoding of the IUPAC names in the training set. The MolFormulas and SELFIES are tokenized on a per-atom basis. Property values are represented as either scalars or decile ranges labeled 1-10 with each digit tokenized separately. Finally, all tags ( `<..>` , `<../>` ) and property names are encoded as special tokens.

### 4.2 TASK DESCRIPTIONS

We evaluate MOLJET on 22 tasks split across 8 different categories: molecular rediscovery, similarity sampling, substructure sampling, isomer generation, median molecules, multi-property optimization, drug-likeness and biological activity. Each task is taken from either the GuacaMol evaluation

---

[1]IUPAC (International Union of Pure and Applied Chemistry) nomenclature provides an international standard of naming compounds which can be used to create unambiguous structural formula.

framework (Brown et al., 2019) or the MIMOSA multi-property optimization framework (Fu et al., 2021). Table 2 provides examples of tasks from a few of the optimization categories and their corresponding prompts. Detailed descriptions of each task category are provided below.

Table 2: Example of the downstream tasks and prompt designs used in the zero-shot evaluation. We color each prompt with the modality(s) that they are associated with. For the prompts for all 22 tasks, please refer to Tables 6 and 7 in Appendix A.

| Task/Example | Prompt |
|---|---|
| Molecular Rediscovery | `<text_type>`**IUPAC**`</text_type>` |
| Celecoxib | `<text>`**4-[5-(4-methylphenyl)..benzenesulfonamide**`</text><mol>` |
| Similarity Sampling | `<text_type>`**FuncGroups**`</text_type>` |
| Albuterol | `<text>`**butylamino,hydroxyethyl,phenol**`</text><mol>` |
| Isomer Generation | `<text_type>`**MolFormula**`</text_type>` |
| $C_{11}H_{24}$ | `<text>`**C11H24**`</text><mol>` |
| Multi-Property Optimization | `<text_type>`**IUPAC**`</text_type>` |
| | `<text>`**N-[2-[2-(dimethylamino)..prop-2-enamide**`</text>` |
| Osimertinib | `<property>`**tpsa**`</property><val>`**146.0**`</val>` |
| | `<property>`**logp**`</property><val>`**-0.5**`</val><mol>` |

**Molecular Rediscovery.** The model must generate an exact match to the target. This task tests the model's ability to explore regions of molecular phase space which it has not encountered during training.

**Similarity Sampling.** The model must generate many samples that are structurally similar to the target but not an exact match. This task tests the model's ability to make small structural modifications to a target without diverting too far from the original molecule. This is analogous to how a chemist might approach the design of a new drug by modifying small chemical motifs of a starting structure to improve a specific desired behavior while maintaining other drug-like qualities from the original molecule.

**Substructure Sampling.** The model must generate many samples that contain a specific structural motif or set of motifs. In some tasks, the model may also be penalized for generating molecules with non-desired motifs or for diverging too far from the pharmacological properties of the molecule from which the desired motif is drawn. This task tests the model's ability to generate functional moieties off a scaffold or "fill in" the scaffold given a set of functional moieties.

**Isomer Generation.** The model must generate as many structural isomers as it can from a given molecular formula. This task tests the model's ability to map coarse-grained chemical information to a fully connected atomic graph. It also tests if the model can enumerate all possible structures from a local region of chemical phase space.

**Median Molecules.** The model must generate samples that are maximally similar to two different target molecules. This task tests the model's ability to interpolate between two valid chemical structures, a common goal when trying to discover a molecule that maximizes the desired properties of two separate existing molecules.

**Multi-Property Optimization (MPO).** The model must simultaneously match both structural and property requirements as dictated by the task. For instance, the model might be tasked with finding a structural analogue to the antihistamine fexofenadine that is "less greasy" by reducing the LogP and increasing the TPSA while maintaining a high structural similarity to the target. These tasks put the model in realistic drug design scenarios and demonstrate its ability to perform structural sampling while also constraining the generated molecules to the desired property ranges.

To demonstrate the versatility of the MOLJET framework, we also evaluate the model on the multi-property optimization tasks outlined in Fu et al. (2021). These require the model to maintain high structural similarity to an input drug-like molecule while simultaneously maximizing PLogP and

either QED (**Drug-Likeness**) or DRD2 (**Biological Activity**). We report performance on these two tasks as success rate which is defined as the proportion of input molecules that the model is able to improve beyond a pre-defined threshold for each property while maintaining high similarity. Further details on the definition of success rate are provided in Jin et al. (2018b). Each GuacaMol task is evaluated based on a weighted average of the top 100 scoring molecules for that task. Further details on the definitions of each GuacaMol metric are provided in Brown et al. (2019) and Appendix E.

**Conditional Language Model Pretraining.** We train two independent version of MOLJET, MOLJET-GUAC and MOLJET-BIO. MOLJET-GUAC is trained and evaluated with the three text types and TPSA, LogP, BertzCT, number of fluorine atoms and ring counts (total and aromatic). MOLJET-BIO is trained and evaluated with the three text types and PLogP, QED and DRD2. We train two additional model variants - one to study the difference between scalar and decile property value representations (MOLJET-GUAC$_{\text{SCALAR/DECILE}}$) and one without property conditioning to study the cumulative effect that additional modalities have on text-only inference tasks (MOLJET-GUAC$_{\text{TEXT-ONLY/TEXT+PROP}}$). The models are pretrained from scratch on the filtered PubChem training set. Further details on the training procedure, hyperparameters, baseline models and sampling scheme can be found in Appendices C & D.

## 5 EXPERIMENTAL RESULTS

The performances of MOLJET-BIO and MOLJET-GUAC on the MIMOSA and GuacaMol evaluation frameworks are displayed in Tables 3 and 4. Both models are very competitive during zero-shot inference with MOLJET-GUAC outperforming $\sim 78\%$ of all baselines on the GuacaMol benchmarks and MOLJET-BIO improving the success rate on the Drug-Likeness and Biological Activity tasks by 18.75% and 13.5% respectively. It should be noted that the baselines are fine-tuned on each task in a supervised manner, whereas MOLJET has only undergone self-supervised pretraining and is seeing the task-specific optimization prompts for the first time during inference. Thus, the performance on these benchmarks demonstrates the efficacy of our multimodal framework in generalizing to previously unseen molecular distributions.

**Multi-Property Optimization.** We first show that MOLJET is able to leverage information from multiple modalities to simultaneously control the structure and properties of generated molecules during *zero-shot* inference. By conditioning the model on the modalities that are optimal for a given task, it can generate molecular distributions that outperform previously state-of-the-art baselines on a variety of multi-property optimization benchmarks. It accomplishes this by inferring how the desired structural features must be modified to satisfy the additional property constraints. We use the conditional generation sampling method described in Section 3.3 to efficiently explore the local region of molecular phase space dictated by the multimodal prompt.

For example, MOLJET-BIO outperforms the previous state-of-the-art, MIMOSA, in both absolute property improvement and success rate on the Drug-Likeness and Biological Activity MPO tasks. It does so by exploring the local region of molecular phase space surrounding the target molecule more efficiently by directly sampling from the conditional distribution. Because MIMOSA makes iterative modifications to the target molecule, it does not venture as far from the original structure during optimization. While this leads to a higher similarity score on both tasks, it fails to find as many molecules that satisfy the property optimization constraints and thus has a lower success rate.

Table 3: Benchmark results on the MIMOSA MPO evaluation framework. PLogP, QED and DRD2 columns refer to the absolute improvement in property values from successful samples.

| Method | Drug-Likeness | | | | Biological Activity | | | |
| --- | --- | --- | --- | --- | --- | --- | --- | --- |
| | Similarity | PLogP | QED | Success | Similarity | PLogP | DRD2 | Success |
| VJTNN | 0.17 | 0.46 | 0.02 | 1.0% | 0.18 | 0.55 | 0.27 | 3.4% |
| DeepGA | 0.35 | 0.93 | 0.09 | 24.9% | 0.38 | 0.68 | 0.20 | 29.3% |
| MIMOSA | **0.42** | 0.93 | 0.10 | 32.0% | **0.54** | 0.75 | 0.35 | 43.7% |
| MOLJET-BIO (Zero-shot) | 0.37 | **1.19** | **0.14** | **38.0%** | 0.35 | **3.38** | **0.48** | **49.6%** |

Table 4: Benchmark results on GuacaMol which contains both MPO and molecular structure generation tasks. Bold values indicate the best performing model and underlined values indicate the second best performing model measured against the baselines.

| Benchmark Category | Best of Data Set | SMILES LSTM | SMILES GA | Graph GA | MOLJET-GUAC (Zero-shot) | MOLJET-GUAC + Graph GA |
|---|---|---|---|---|---|---|
| MPOs | 0.698 | 0.778 | 0.717 | 0.868 | 0.838 | **0.878** |
| Rediscovery | 0.613 | **1.000** | 0.523 | 0.945 | **1.000** | **1.000** |
| Similarity | 0.546 | **1.000** | 0.771 | 0.977 | **1.000** | **1.000** |
| Substructure | 0.643 | 0.973 | 0.769 | **0.985** | 0.817 | **0.985** |
| Isomers | 0.716 | 0.912 | 0.745 | 0.954 | **1.000** | **1.000** |
| Median | 0.371 | 0.403 | 0.362 | 0.417 | 0.409 | **0.447** |
| Total | 0.623 | 0.850 | 0.671 | 0.877 | 0.857 | **0.900** |

We observe a similar trend from *zero-shot* MOLJET-GUAC on the GuacaMol MPOs. When breaking the tasks down individually, it outperforms all three baselines on the ranolazine, perindopril, and amlodipine MPOs and is within 1% and 2.5% of the best performing model on the fexofenadine and osimertinib MPOs, respectively (Appendix E). These tasks also require the model to meet one or more property specifications while maintaining high similarity to a target molecule (see Fexofenadine and Perindopril MPOs, Figure 2). In total, MOLJET outperforms or is competitive with the leading baseline on seven out of nine MPOs across both evaluation frameworks demonstrating the versatility and efficacy of our multimodal framework.

**Conditional Molecular Structure Generation.** MOLJET-GUAC also performs well at the *zero-shot* molecular structure generation tasks, achieving a perfect score on rediscovery, similarity sampling and isomer generation (Table 4). This indicates that the model is able to accurately estimate the molecular structural probability manifold of the training set and navigate it based on the conditional multimodal prompts. Each of the three text modalities provide a different degree of structural specificity with which the model can be conditioned. For instance, tasks with stringent similarity requirements are better suited for IUPAC conditioning, whereas FuncGroup conditioning yields a more diverse set of generated molecules (see Drug-Likeness vs. Fexofenadine MPO in Fig. 2). FuncGroup conditioning is also the most flexible as it can be used to combine the structural characteristics of multiple input molecules (see Median Molecules, Fig. 2).

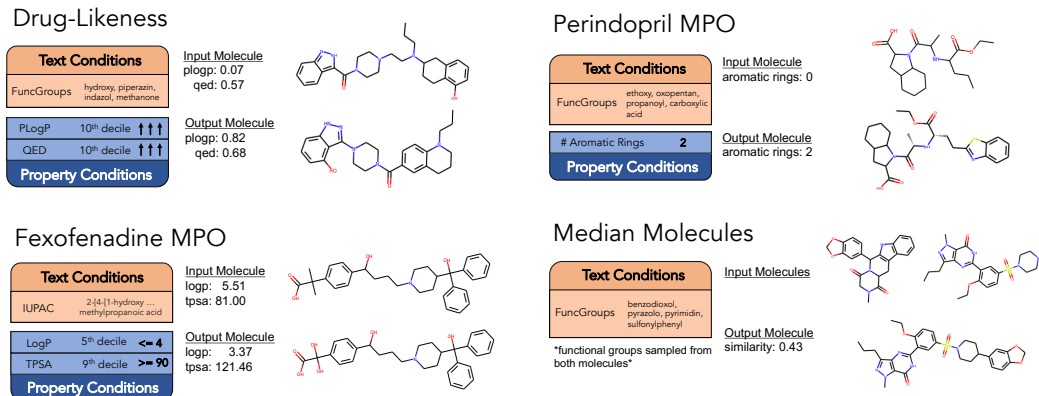

Figure 2: Prompts, inputs and high-scoring samples for four of the *de novo* design tasks.

We confirm these observations quantitatively by measuring the performance of each text modality individually on the similarity sampling tasks. We choose similarity as it is the most common structural objective for the MPOs and thus highlights important differences in sampling performance for realistic drug design scenarios. The results of this experiment are shown in Figure 3. As expected, we explore the largest subset of relevant phase space when conditioning on FuncGroups. How-

ever, there are some circumstances where IUPAC conditioning is just as effective, namely when the molecule is complex such as the stereoisomer mestranol.

To estimate how amenable MOLJET is to further optimization, we re-run the Graph GA method but replace the starting population with the top 100 molecules generated by MOLJET. On average, the Graph GA seeded with molecules generated by MOLJET improves upon the zero-shot MOLJET by $\sim 5\%$ and the baseline Graph GA by $\sim 2.6\%$ (Table 4). This demonstrates the capacity of MOLJET to be further improved by task-specific fine-tuning strategies and we leave further work in this direction as future research.

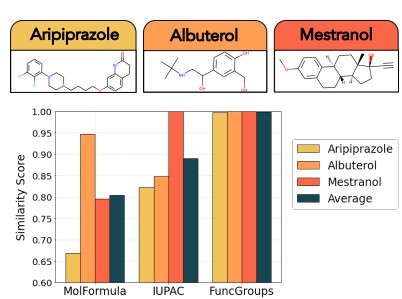

Figure 3: Similarity sampling from each text modality.

**Evaluating Prompt Design.** We also run ablations to study a) the effect of the choice of numerical property representation on the GuacaMol tasks with property conditioning and b) the impact of the inclusion of property modalities during training on GuacaMol tasks with text-only conditioning. On the GuacaMol tasks with property conditioning, MOLJET-GUAC$_{\text{SCALAR}}$ performs slightly better than MOLJET-GUAC$_{\text{DECILE}}$ (0.881 vs. 0.872). This suggests that the property prediction capacity of the scalar model is only slightly greater than the average distance between decile bins. For most properties, this distance is fairly large so this result indicates a potential area in which MOLJET could be improved.

Finally, we evaluate MOLJET-GUAC$_{\text{TEXT-ONLY}}$ and MOLJET-GUAC$_{\text{TEXT+PROP}}$ on the text-only inference tasks from GuacaMol (Table 5). These tasks do not require any property conditioning during inference and thus the performance of the two models should be expected to be comparable if cross-modal learning does not occur during training. However, we find that MOLJET-GUAC$_{\text{TEXT+PROP}}$ performs *better* on the text-only inference tasks, supporting our hypothesis that our multimodal prompt design framework supports both inter- *and* cross-modal learning. The property information that is jointly embedded during training enhances the models understanding of molecular structure even when that information is not provided during inference.

Table 5: Multimodal Model Ablations

| Modality | GuacaMol | Reconstruction | |
| --- | --- | --- | --- |
| | | IUPAC | FuncGroup |
| Text | 0.827 | 62.1% | 60.2% |
| Text + Property | **0.843** | **68.7%** | **63.4%** |

To confirm this behavior, we construct two additional text-only inference tasks, **IUPAC Reconstruction** and **FuncGroup Reconstruction**. IUPAC Reconstruction tests the models ability to accurately reconstruct a SELFIES string given its IUPAC from a holdout set of IUPAC-SELFIES pairs that were not seen during training. FuncGroup Reconstruction tests the models ability to generate molecules that contain the requested functional group from a list of 102 functional groups developed by the authors to include a wide range of atom types and complexities. Additional implementation details for each task are outlined in Appendices A & F. Again, we find that MOLJET-GUAC$_{\text{TEXT+PROP}}$ outperforms MOLJET-GUAC$_{\text{TEXT-ONLY}}$, providing additional evidence that both inter- and cross-modal learning occur during training and that multimodal joint embeddings are capable of enhancing the performance of *de novo* molecular design models

# 6 CONCLUSION

We introduce MOLJET, a multimodal foundational chemistry model for conditional *de novo* design of organic molecules. MOLJET demonstrates state-of-the-art performance on realistic drug design tasks in a zero-shot manner. Our framework is adaptable and easy to interpret, making it well-suited for the inclusion of other modalities such as scientific text. We make our code, models and data publicly available and provide API access to our pretrained models to allow chemistry researchers of all backgrounds to participate in the future development of AI-driven *de novo* molecular design.

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

# A  PROMPT DESIGN

Table 6: Example of the multi property optimization tasks and prompt designs used in the zero-shot evaluation. We color each prompt with the modality(s) that they are associated with.

| Example | Prompt |
|---|---|
| Osimertinib | `<text_type>`**IUPAC**`</text_type>`
`<text>`**N-[2-[2-(dimethylamino)..prop-2-enamide**`</text>`
`<property>`**tpsa**`</property><val>`**146.0**`</val>`
`<property>`**logp**`</property><val>`**-0.5**`</val><mol>` |
| Fexofenadine | `<text_type>`**IUPAC**`</text_type>`
`<text>`**2-[4-(1-hydroxy...methylpropanoic acid]**`</text>`
`<property>`**tpsa**`</property><val>`**9**`</val>`
`<property>`**logp**`</property><val>`**5**`</val><mol>` |
| Ranolazine | `<text_type>`**IUPAC**`</text_type>`
`<text>`**N-(2,6-dimethylphenyl...piperazin-1-yl]acetamide**`</text>`
`<property>`**logp**`</property><val>`**8.5**`</val>`
`<property>`**aromatic_rings**`</property><val>`**0**`</val><mol>`
`<property>`**f_count**`</property><val>`**1**`</val><mol>` |
| Perindopril | `<text_type>`**FuncGroups**`</text_type>`
`<text>`**ethoxy,oxopentan,octahydroindole,carboxylic acid**`</text>`
`<property>`**aromtic_rings**`</property><val>`**2**`</val>` |
| Amlodipine | `<text_type>`**FuncGroups**`</text_type>`
`<text>`**aminoethoxymethyl,chlorophenyl,dihydropyridine,dicarboxylate**`</text>`
`<property>`**ring_count**`</property><val>`**3**`</val>` |
| Sitagliptin | `<text_type>`**FuncGroups**`</text_type>`
`<text>`**amino,trifluoromethyl,triazolo,pyrazin**`</text>`
`<text_type>`**MolFormula**`</text_type>`
`<text>`**C16H15F6N5O**`</text>`
`<property>`**logp**`</property><val>`**3**`</val>`
`<property>`**tpsa**`</property><val>`**6**`</val><mol>` |
| Zaleplon | `<text_type>`**IUPAC**`</text_type>`
`<text>`**N-[3-(3-cyanopyrazolo...N-ethylacetamide]**`</text>`
`<text_type>`**MolFormula**`</text_type>`
`<text>`**C19H17N3O2**`</text>` |
| PLogP/QED

*(Drug-Likeness)* | `<text_type>`**FuncGroups**`</text_type>`
`<text>`**oxo,phenyl,triazaspiro,indole,carboxamide**`</text>`
`<property>`**plogp**`</property><val>`**10**`</val>`
`<property>`**qed**`</property><val>`**10**`</val><mol>` |
| PLogP/DRD2

*(Biological Activity)* | `<text_type>`**FuncGroups**`</text_type>`
`<text>`**oxo,triazolo,methoxyethyl,benzimidazol,dimethylacetamide**`</text>`
`<property>`**plogp**`</property><val>`**10**`</val>`
`<property>`**drd2**`</property><val>`**10**`</val><mol>` |

Table 7: Example of the conditional molecular structure generation tasks and prompt designs used in the zero-shot evaluation. We color each prompt with the modality(s) that they are associated with.

| Task | | Example | Prompt |
|---|---|---|---|
| **Molecular** | **Rediscovery** | Celecoxib | `<text_type>`**IUPAC**`</text_type>`
`<text>`**4-[5-(4-methylphenyl)..benzenesulfonamide**`</text><mol>` |
| | | Troglitazone | `<text_type>`**IUPAC**`</text_type>`
`<text>`**5-[[4-[(6-hydroxy...thiazolidine-2,4-dione]]]**`</text><mol>` |
| | | Thiothixene | `<text_type>`**IUPAC**`</text_type>`
`<text>`**(9Z)-N,N-dimethyl...thioxanthene-2-sulfonamide**`</text><mol>` |
| **Similarity** | **Sampling** | Albuterol | `<text_type>`**FuncGroups**`</text_type>`
`<text>`**butylamino,hydroxyethyl,phenol**`</text><mol>` |
| | | Aripiprazole | `<text_type>`**FuncGroups**`</text_type>`
`<text>`**dichlorophenyl,piperazin,quinolin**`</text><mol>` |
| | | Mestranol | `<text_type>`**FuncGroups**`</text_type>`
`<text>`**ethynyl,methoxy,methyl,octahydro,phenanthren**`</text><mol>` |
| **Isomer** | **Generation** | $C_{11}H_{24}$ | `<text_type>`**MolFormula**`</text_type>`
`<text>`**C11H24**`</text><mol>` |
| | | $C_9H_{10}N_2O_2PF_2Cl$ | `<text_type>`**MolFormula**`</text_type>`
`<text>`**C9H10N2O2PF2Cl**`</text><mol>` |
| **Median** | **Molecules** | Camphor/Menthol | `<text_type>`**FuncGroups**`</text_type>`
`<text>`**heptan,methyl,trimethylbicyclo,ylcyclohexan**`</text><mol>` |
| | | Tadalafil/Sildenafil | `<text_type>`**FuncGroups**`</text_type>`
`<text>`**pyrazolo,triazatetracyclo,pyrimidin,methylpiperazin**`</text><mol>` |
| **Substructure** | **Sampling** | Valsartan | `<text_type>`**IUPAC**`</text_type>`
`<text>`**methanoyl-methyl...phenyl]methyl]amine**`</text><mol>`
`<property>`**logp**`</property><val>`**2.0**`</val><`
`<property>`**tpsa**`</property><val>`**77.0**`</val><`
`<property>`**bertzct**`</property><val>`**896.4**`</val><` |
| | | Deco Hop | `<text_type>`**FuncGroups**`</text_type>`
`<text>`**amino,hydroxy,quinazoline**`</text><mol>` |
| | | Scaffold Hop | `<text_type>`**FuncGroups**`</text_type>`
`<text>`**propanol,benzothiazol**`</text><mol>` |

## B    PROMPT SAMPLING STRATEGY

Prompts are stochastically generated from the available modalities by the following set of rules:

- The text modality is sampled uniformly from the list `(IUPAC, FuncGroups, MolFormula, None)`. If `None` is selected then no text conditioning is included for that sample. This allows the user to perform property-only conditioning by leaving out the text conditioning during inference.
- If `FuncGroups` is chosen, then the number of functional groups, N, used for conditioning is sampled uniformly from `[1-M]` where M is the total number of functional groups for the given molecule. Then N functional groups are selected from the list and concatenated with commas.
- Next, the number of property conditions, K, is sampled uniformly from `[0-L]` where L is the total number of property modalities available for training. Then K properties are chosen from the list and their property names and values are added to the prompt after the text type and text. The ordering of property sub-modalities is also stochastic.

## C    TRAINING & SAMPLING IMPLEMENTATION DETAILS

We use the GPT-NeoX Python library Andonian et al. (2021) developed with Megatron Shoeybi et al. (2019) and DeepSpeed Rasley et al. (2020). We optimize the autoregressive log-likelihood (*i.e.,* cross-entropy loss) averaged over a 256-token context. We set the global batch size as 2048, and the learning rate to $2 \times 10^{-4}$, and rely on the cosine decay. We use an Adam optimizer with $\beta_1 = 0.9$, $\beta_2 = 0.99$, and $\sigma = 10^{-8}$ and clip the gradient norm at 1.0. We use the Rotary positional embeddings Su et al. (2021), parallel attention and feed-forward (FF) Black et al. (2022), and all dense layers in comparison to the original transformer decoder model architecture Radford et al. (2019).

We use a $q$ temperature value of 1.0 for sampling for evaluating all 22 tasks. We found that this value gives us the best tradeoff between the validity and diversity of the generated molecules. For each GuacaMol task, we generate 128K samples to use for evaluation. This is on the order of the number of samples that are generated and evaluated during fine-tuning of the GuacaMol baselines. For the Drug-Likeness and Biological Activity tasks, we evaluate on 250 molecules randomly sampled from a subset of the ZINC dataset provided in Jin et al. (2018b) in accordance with the methods outlined in Fu et al. (2021). For each molecule, we generate 1K samples which is on the order of the number of samples that are generated and evaluated during fine-tuning of the MIMOSA baselines.

## D    BASELINE MODELS

We compare MOLJET to two sets of baselines – one for the GuacaMol tasks and another for the Drug-Likeness/Biological Activity tasks. The GuacaMol baselines include:

- **Best of Data Set**, the metrics evaluated on the top molecules from the ChEMBL dataset (Gaulton et al., 2012))
- **SMILES LSTM**, an LSTM model which is fine-tuned with the hill-climbing method (Brown et al., 2019))
- **SMILES GA**, a genetic algorithm that makes mutations to a SMILES string (Yoshikawa et al., 2018))
- **Graph GA**, a genetic algorithm that makes mutations directly to a molecular graph (Jensen, 2019))

The Drug-Likeness/Biological Activity baselines include:

- **VJTNN**, a graph-to-graph translation VAE that utilizes adversarial regularization (Jin et al., 2018b))
- **DeepGA**, a genetic algorithm enhanced with a discriminator neural network to improve molecular diversity (Nigam et al., 2019))

- **MIMOSA**, a Markov chain Monte Carlo sampling strategy augmented by pretrained graph neural networks (Fu et al., 2021))

# E    MODEL PERFORMANCE ON INVIDIDUAL GUACAMOL TASKS

Table 8 shows the detailed performance view on the GuacaMol benchmark. Aside from the rediscovery tasks, the final score for each metric is evaluated as a weighted average of the top 100 scoring molecules that were generated during sampling. The scores for individual molecules are based on their ECFP4 (Rogers & Hahn, 2010) fingerprint similarities to the targets, calculated property values and structural features. These values are passed through a set of modifiers and thresholds to scale them between 0 and 1. The score is then calculated as the geometric mean of each scaled task-specific value. For further details on the metric definition of each benchmark, please refer to Brown et al. (2019).

Table 8: Benchmark results on GuacaMol which contains both MPO and molecular structure generation tasks. Bold values indicate the best performing model and underlined values indicate the second best performing model

| Benchmark Category | Benchmark | Best of Data Set | SMILES LSTM | SMILES GA | Graph GA | MOLJET-GUAC (Zero-shot) | MOLJET-GUAC + Graph GA |
|---|---|---|---|---|---|---|---|
| MPOs | Osimertinib | 0.781 | 0.894 | 0.880 | 0.937 | 0.914 | **0.992** |
| | Fexofenadine | 0.817 | 0.926 | 0.904 | **1.000** | 0.997 | **1.000** |
| | Ranolazine | 0.836 | 0.833 | 0.832 | 0.913 | **0.920** | **0.920** |
| | Perindopril | 0.701 | 0.764 | 0.644 | 0.803 | **0.804** | 0.823 |
| | Amlodipine | 0.696 | 0.885 | 0.678 | 0.888 | **0.895** | 0.903 |
| | Sitagliptin | 0.509 | 0.536 | 0.526 | 0.809 | 0.758 | **0.823** |
| | Zaleplon | 0.547 | 0.610 | 0.552 | **0.728** | 0.625 | 0.688 |
| Rediscovery | Celecoxib | 0.674 | **1.000** | 0.570 | 0.836 | **1.000** | **1.000** |
| | Troglitazone | 0.558 | **1.000** | 0.523 | **1.000** | **1.000** | **1.000** |
| | Thiothixene | 0.608 | **1.000** | 0.476 | **1.000** | **1.000** | **1.000** |
| Similarity | Albuterol | 0.522 | **1.000** | 0.871 | **1.000** | **1.000** | **1.000** |
| | Aripiprazole | 0.595 | **1.000** | 0.747 | 0.985 | 0.999 | **1.000** |
| | Mestranol | 0.520 | **1.000** | 0.695 | 0.945 | **1.000** | **1.000** |
| Substructures | Valsartan | 0.259 | 0.931 | 0.628 | 0.958 | 0.930 | **0.977** |
| | Deco Hop | 0.933 | **0.996** | 0.876 | 0.995 | 0.893 | **0.996** |
| | Scaffold Hop | 0.738 | 0.993 | 0.803 | **1.000** | 0.632 | 0.984 |
| Isomers | $C_{11}H_{24}$ | 0.684 | 0.963 | 0.734 | 0.952 | **1.000** | **1.000** |
| | $C_9H_{10}N_2O_2PF_2Cl$ | 0.747 | 0.860 | 0.757 | 0.955 | **1.000** | **1.000** |
| Median | Camphor/Menthol | 0.334 | 0.398 | 0.348 | 0.405 | 0.386 | **0.416** |
| | Tadalafil/Sildenafil | 0.407 | 0.408 | 0.377 | 0.429 | **0.434** | 0.478 |
| Total | — | 0.623 | 0.850 | 0.671 | 0.877 | 0.857 | **0.900** |

# F    RECONSTRUCTION TASKS

To validate the ablation on the Text + Property vs. the Text-Only models, we construct two additional tasks that evaluate the model's performance on text-only conditioning - IUPAC Reconstruction and FuncGroup Reconstruction. An IUPAC reconstruction is counted as successful if the generated SELFIES string exactly matches the canonical SMILES from the holdout set after being decoded back into a SMILES and canonicalized. IUPAC Reconstruction is evaluated on 10000 randomly sampled IUPAC/SMILES pairs from the holdout validation set. A FuncGroup reconstruction is counted as successful when the SMILES string decoded from the generated SELFIES string matches the substructure pattern matching the requested functional group (we use SMARTS substructures for matching). We hand select 102 functional groups to test the model on its ability to recognize simple functional groups, basic nitrogen heterocycles, basic oxygen heterocycles, basic mixed heterocycles, double ring nitrogen heterocycles, double ring oxygen heterocycles, polycyclic aromatic hydrocarbons, fused rings and phenyls among others. The full dataset will be made available upon request.

