# OpenReview forum: "MolJET: Multimodal Joint Embedding Transformer for Conditional de novo Molecular Design and Multi-Property Optimization"
_ICLR.cc/2023/Conference — Submitted to ICLR 2023_

### Official Review · Reviewer_cdBp · 2022-10-24

**Confidence:** 3
**Correctness:** 3
**Technical Novelty And Significance:** 3
**Empirical Novelty And Significance:** 1
**Recommendation:** 3

**Clarity, Quality, Novelty And Reproducibility:**

Clarity: paper is generally clearly written

Novelty/originality: I think the prompt engineering idea in this paper is novel, but the general idea is very related to a lot of existing work on molecular transformers. I would say that this paper's contribution is only slightly novel.

Quality: overall the quality is ok. To me the main thing that is lacking is the experimental evaluation.

Reproducibility: not great since code is not included, but the prompt engineering is described well so I imagine a reader could reimplement it.

**Strength And Weaknesses:**

Strengths: reasonable idea (the same sort of thing worked well in language, so why not chemistry?), clear writing, good ideas for prompt generation

Weaknesses:

- Missed a lot of related work. There have been many papers proposing "foundation models" for molecules [1-5], but the discussion is minimal. My understanding is that most of these methods have essentially tried something very similar to this work. The key difference between this work and previous works seems to be the structure of the prompt, but it would be good for the authors to acknowledge the previous work and explain more clearly how their contribution expands upon previous work.
- Missed related work on optimizing molecules: there have been many papers focused on producing molecules with optimized properties, but the authors do not discuss this or cite many of them. I think the authors' focus in the related work section was too narrow by only focusing on a handful of works which explicitly focus on multi-property optimization. A common method of multi-property optimization is to combine all the objectives together into a single scalar objective, then use a single task optimization technique (see [6] which benchmarks many of these methods). I think this needs to be discussed more to explain how this work fits into a large existing literature on property optimization.
- Method seems geared towards trivial properties and unlikely to work for more complicated real-world properties. The authors train the transformer on prompts constructed from easily-available molecular properties like QED, ring counts, TPSA (things all available from `rdkit`). The benchmarks from GuacaMol all involve producing molecules with specific values of these properties, so it is not surprising that it does well on them. However:
  1. Producing molecules like this is not a challenging problem since these properties can easily be calculated on a computer. Therefore one-shot generation isn't necessary or desired; instead iterative algorithms like those in GuacaMol could be used.
  2. The thing which _is_ desired is to produce molecules with non-trivial properties (e.g. binding affinities, toxicities) given few (or no) labelled training examples. However, the authors do not show this, and it seems incredibly unlikely that MolJET would successfully do this without a large quantity of training data. The authors do experiments on DRD2, but as far as I can tell this is the output of a predictive model, not an experimental measurement (although the authors should clarify this). Therefore this method does not seem to be posed to solve practical, real-world problems.
- The origin of the numbers in Table 3-4 are unclear. Are these averages over multiple tasks? There should be error bars. I think not reporting error bars is bad practice.
- One critical detail which the authors only mention in the appendix is that the numbers in Tables 3-4 come from the best of _128,000_ samples from the model (i.e. they are the top 0.1% of all outputs). I think it is not appropriate to call this "zero-shot": in real life one would need to evaluate these 100k+ proposed molecules to find the best one, which creates a lot of training data in the process. Therefore it would be more appropriate to use an iterative optimization algorithm. I think it would be more honest to plot the distribution of property scores from the molecules outputted by MolJET.
- The authors only used baselines present in the original GuacaMol paper. Since its publications other methods have beaten these scores, for example [7]. The table from [7] also contains many other baseline methods. I think the authors should use it.

[1] Rong, Yu, et al. "Self-supervised graph transformer on large-scale molecular data." Advances in Neural Information Processing Systems 33 (2020): 12559-12571.

[2] Zhang, Zaixi, et al. "Motif-based graph self-supervised learning for molecular property prediction." Advances in Neural Information Processing Systems 34 (2021): 15870-15882.

[3] Chithrananda, Seyone, Gabriel Grand, and Bharath Ramsundar. "ChemBERTa: large-scale self-supervised pretraining for molecular property prediction." arXiv preprint arXiv:2010.09885 (2020).

[4] Wang, Sheng, et al. "SMILES-BERT: large scale unsupervised pre-training for molecular property prediction." Proceedings of the 10th ACM international conference on bioinformatics, computational biology and health informatics. 2019.

[5] Irwin, Ross, et al. "Chemformer: a pre-trained transformer for computational chemistry." Machine Learning: Science and Technology 3.1 (2022): 015022.

[6] Gao, Wenhao, et al. "Sample Efficiency Matters: A Benchmark for Practical Molecular Optimization." Thirty-sixth Conference on Neural Information Processing Systems Datasets and Benchmarks Track.

[7] Ahn, Sungsoo, et al. "Guiding deep molecular optimization with genetic exploration." Advances in neural information processing systems 33 (2020): 12008-12021.

**Summary Of The Paper:**

This paper proposes to train a "foundation" transformer model on inputs featuring molecule names, properties, and their string representations. The transformer can then be used for optimization by prompting it with molecular names and properties, then predicting the most likely molecule to result. They perform several experiments on GuacaMol tasks (toyish tasks) with promising results.

**Summary Of The Review:**

The technique proposed by this paper uses interesting prompt engineering, but seems unlikely to be useful for real-world problems and is very similar to other work on transformers for molecules. I also did not find their experimental results convincing. Therefore I recommend rejection of the current manuscript.

---

### Official Review · Reviewer_qqLn · 2022-10-25

**Confidence:** 3
**Correctness:** 2
**Technical Novelty And Significance:** 2
**Empirical Novelty And Significance:** 2
**Recommendation:** 3

**Clarity, Quality, Novelty And Reproducibility:**

The paper is lacking detail on the tuning of the method and lacks explicit code or data, so the reproducibility status and the quality are unclear.  The novelty of the work is modest and could improve with additional work.
The paper does not make a clear case on where this method would be useful, especially given the plethora of foundation models trained on other properties.


**Strength And Weaknesses:**


The high level idea behind this model originates from conditional generation methods in natural language, and has some promise.  The paper demonstrates the method on two benchmarks.

The paper only includes a limited review of relevant literature; it does not highlight existing promising mixed representations of small molecules that include the 2D graphs or 3D coordinates and that have shown a level of success in mixed models.  Unfortunately the paper does not provide code, explicit data, or descriptions of the framework or hardware used in training, so it is hard to verify the correctness or assess the reproducibility of the method presented in this paper.  It is not clear a priori to me if the ECFP4 similarity of 0.343 for holdout removes  memorized solutions from these benchmarks when genetic algorithms also come into play in Table 4, which could easily recombine the benchmark molecules from prior pieces and predict properties as linear combinations. I like and encourage the author's sentiment for going after big models, however, it would be nice to pick a downstream task that would be impossible to trivially guess from the training (for example finetuning to the prediction of a new property not included in the training set; for example, take a look at the open graph benchmarks, including the large-scale challenge for small molecules.)  I have a pet peeve against the optimization of the property PLogP, which I believe is unlikely to come up in a real drug discovery project (who wants to make their molecules more hydrophobic and more linear?)---do the authors think that being able to optimize this property adds value to molecule generators, and if so, why?  The listed DRD2 success rate in table 3 is small compared to the graph-to-graph applications on the same problem, so I'd recommend that the authors do a more extensive review and update the table with the correct state of the art results. Finally, why is the reconstruction from IUPAC in Table 5 so low?  Wouldn't the pretraining include translation and didn't past translation models from IUPAC to SELFIES basically achieve perfect results?



**Summary Of The Paper:**

This paper describes a large transformer model that generates small molecule SELFIES from input prompts that convey properties of relevance in molecular design. The model shows good performance on GuacaMol and MIMOSA, two previously published benchmarks.


**Summary Of The Review:**


Overall, I believe that this work is more appropriate for a specialized audience and perhaps it could benefit from additional discussions between the authors and those specialists, for example about how to design an improved benchmark to demonstrate the power of such a pretrained model.  The current draft is *not* strong enough and broad enough for publication in ICLR2023.

---

### Official Review · Reviewer_TDmG · 2022-10-29

**Confidence:** 5
**Correctness:** 2
**Technical Novelty And Significance:** 2
**Empirical Novelty And Significance:** 2
**Recommendation:** 3

**Clarity, Quality, Novelty And Reproducibility:**

* The majority of this paper is clear and easy to follow. However, my main concern is still why we would introduce text, one merit I see is human-readable, but from its current form, it still needs to define properties during training which is supervision anyways.
* I consider the introduction of prompt and text-based foundation model to molecule generation as somewhat novel.
* Despite the authors claim in the conclusion, the code, models, and data are publicly available, but no anonymous link or supplementary material is attached to this submission. Therefore, reproducibility cannot be 100% guaranteed.

**Strength And Weaknesses:**

Strength
* There is certainly merit to adopt a prompt-based method from the language model for flexible molecule generation.
* The developed method is simple and technically sound.
* The experiments are conducted on reliable benchmarks (e.g. GuacaMol).

Weaknesses
* Even though the " multimodal foundation model" for molecule generation sounds desirable, I do not see it is well-supported or has empirical merit of it yet. First of all, the proposed model is trained over 100M molecule dataset. Compared to the baseline methods, tons of knowledge, or at least molecular structures and simple properties such as QED and logP, are integrated into the model. It is hard to tell the merit of such a multimodal foundation model over the other compared methods.
* Is this multimodal foundation model suitable for the predictive task?
* This paper misses a large number of important literature on molecule generation [2], even though the authors argue another modality such as graph is left for future study, again this lowers the significance to introduce a text-based model if not compared against other approaches. Even for the string-based method, [1] is missing.
* Related to the previous point, it is not compared against many other molecule optimization methods, even though the authors may argue they focus on single property optimization. I would suggest the authors add a single property optimization task as a comparison to other methods.
* Even though it is called zero-shot generation, I feel this is a bit misleading since given the pretraining context, it is almost equivalent to train a predictor for properties.

[1] Flam-Shepherd, D., Zhu, K. and Aspuru-Guzik, A., 2022. Language models can learn complex molecular distributions. Nature Communications, 13(1), pp.1-10.
[2] Du, Y., Fu, T., Sun, J. and Liu, S., 2022. MolGenSurvey: A Systematic Survey in Machine Learning Models for Molecule Design. arXiv preprint arXiv:2203.14500.

**Summary Of The Paper:**

This paper proposes a multimodal joint embedding transformer for molecule generation. Specifically, it leverages both text descriptions, molecular string descriptors, and molecular properties to train a join embedding and introduces prompt building on the success of language models. After training such a model, the user can use prompt for a variety of molecule generation tasks, such as property optimization, isomer generation, etc.

**Summary Of The Review:**

Overall, I think this is a good initial attempt to introduce text-based pretraining to molecule generation. However, from its current form, I do not strongly agree that it is a significant contribution yet mainly because of its problem formulation, motivation and experiment results. I would put a high standard on this since it is claimed the first or at least from what I know the first batch of works in this direction, it should be well-motivated and formulated for other people to follow in the future. I am happy to adjust my score if the authors make a good rebuttal.

---

### Official Review · Reviewer_KXSs · 2022-10-31

**Confidence:** 4
**Correctness:** 4
**Technical Novelty And Significance:** 3
**Empirical Novelty And Significance:** 3
**Recommendation:** 8

**Clarity, Quality, Novelty And Reproducibility:**

The idea in this paper can be viewed as an exploration of the unification models developed from NLP and computer vision in molecular design. In the context of molecular science, the text-based prompt input for modality fusion can be viewed as the major novelty in this work. This paper can be better presented if input example can be given exactly for each task. Hyperparameters for the model and training process should be provided in this paper.

**Strength And Weaknesses:**

Strengths:
1. One major strengths of this work is text-based prompt design for multimodal fusion within the Transformer model, such that the model can do versatial tasks.
2. The model MolJET can be used for various conditional design and multi-property optimization tasks. Experimental results show that MolJET can achieve excellent tasks.

Weaknesses:
1. Math expressions can be better formulated. For example, in Eq. 1, x is at the left-hand side, however s appears at eh right-hand side. In Eq. 2, i is not reflected in the sum. log -> \log, etc.
2. Minors: transformer -> Transformer, tanimoto -> Tanimoto.

**Summary Of The Paper:**

The authors of this paper investigate a unified model for multiple tasks in molecular science. The model named MolJET is based on Transformer and inspired by unification models in NLP. MolJET takes a prompt sequence and is able to conduct multiple tasks. Experimental results on several tasks show better or competitive performance compared with baselines.

**Summary Of The Review:**

Overall, it is a good paper in the area of molecular design and optimization. This proposed model is able to conduct multiple tasks in zero-shot settings.

---

### Official Review · Reviewer_rKPy · 2022-11-01

**Confidence:** 3
**Correctness:** 3
**Technical Novelty And Significance:** 4
**Empirical Novelty And Significance:** 4
**Recommendation:** 3

**Clarity, Quality, Novelty And Reproducibility:**

- During training, what value do you fill if any prompt is missing, or just leave it blank with only tags?
- How do you decide what modalities to be included in the model pretraining and what is the motivation to train two versions. Is it possible to pre-train a model including all modalities?
- Expand and cite the SELFIES at its first appearance for audience who is not familiar.

**Strength And Weaknesses:**

*Strength*
- The paper is well written with clear organization.
- The design is really novel to bridge the cross-area research to bring AI to real-world problem
- The experiments are thorough

*Weaknesses*
- minor, see clarity section

**Summary Of The Paper:**

This paper illustrates a new generative foundation model for multi-property optimization in molecular de novo design. The authors leverages transformer based self-supervised learning to pre-train the model with carefully designed input sequence. The input token sequence contains multiple prompts which are human interpretable. The model is able to conduct conditional generation in a zero-shot manner. The model is evaluated against an exhaustive list to evaluation tasks. The results show the model ourperforms 80% of the task-specific optimization in zero-shot and achieve SOTA when finetuning. Extensive ablation study also show the necessity of the multimodal design where property improved the performance compared to text-only model. The pretrained model was provided as foundation model for finetuning

**Summary Of The Review:**

The paper is technically sound, and clearly written. The novelty is sufficient to bring multimodal into the AI driven molecular design. The prompt design is smart to handle multiple modalities in the pretraining. Though I am not an expert in this direction, I recommend an accept since the paper is self-contained for easy reading.

---

### Official Review · Reviewer_X1nX · 2022-11-07

**Confidence:** 4
**Correctness:** 3
**Technical Novelty And Significance:** 2
**Empirical Novelty And Significance:** 3
**Recommendation:** 3

**Clarity, Quality, Novelty And Reproducibility:**

The paper is clear and well-written, and the authors will make the model fully available. While the model architecture is very standard, the prompt engineering is novel.


**Strength And Weaknesses:**

Strengths
* A new multimodal "foundation" model for drug-like molecules, with a range of carefully designed prompts. This the first attempt of this specific kind in the area of molecular generation with MPO.
* This model is to be publicly available via an API.
* The paper is clearly written and accessible to a range of audiences, which is important in a cross-disciplinary piece of work.

Weaknesses:
* The experimental results are not strongly convincing. MolJET-GUAC is only outperforming methods not based on MolJET in the isomers task alone. However, Table 4 is ultimately misleading. The best results are in fact obtained by taking MolJET and using it to seed GA. Given that seeding has such a strong impact on the outcome of GA, it is not concrete to claim that the outcome is due to MolJET. To properly substantiate this claim, a more detailed study of the performance of GA under different seeding strategies is a necessary comparison. It is not clear whether the GA-only model was randomly seeded, seeded by similar models or another strategy. It is also notable that MolJET by itself is unable to produce these results, hinting possibly at limited diversity in the molecules sampled from MolJET.
* While the model has been trained to beat standard benchmarks, it is not clearly motivated and explored in more genuine zero-shot settings; many of the properties optimised for are "global" and dependent on a set of substructures being present, with the conditional generation most likely to ensure that the backbone scaffold is in the correct region of chemical space. Therefore, it is not clear that this model has learned any deeper representation of molecules other than one able to recognise functional groups. It would be helpful to include evaluation on more local properties such as binding affinity to specific proteins.
* It is arguable that this represents true "zero-shot" generation. The properties have been included in the training set, and while the holdout set requests molecules optimised for those properties in very different regions of chemical space, the features required to optimize these properties will often be conserved across chemical space. Therefore, performing well on MPO tasks does not represent zero-shot generalization.


**Summary Of The Paper:**

This paper proposes a multimodal transformer for molecular generation with multiple property optimization against multiple objectives, using language-based inputs. The authors describe prompt engineering based upon combinations of descriptive text and string representations of molecular properties and molecules as SELFIES strings. Improved performance on some standard benchmarks for molecular generation, GuacaMol and MIMOSA, are demonstrated.

**Summary Of The Review:**

A multimodal foundation model for MPO is a good idea, and potentially very impactful. In addition, here a strictly text-based modality here makes for a neat and simple model. The paper is clearly written, and in some cases the results on benchmarks appear quite impressive. However, the experimental results are overall not strong as much of the improvement was gained by using the model to seed a genetic optimization algorithm, and this behaviour needs to be explored more thoroughly. The motivations for the particular choice of prompts (apart from the limitations of publicly available data) are not clear and the paper would benefit from more discussion of the likely generalization ability of such a model, and how well it is likely to have learned chemical rules, rather than just substructure matching.

Overall, while the idea is a good one, this is not yet a suitably good contribution because the execution and experimental results are not sufficiently convincing.

---

### Decision · Program_Chairs · 2023-01-20

**Decision:**

Reject

**Justification For Why Not Higher Score:**

Major limitation outweigh the strength.

**Justification For Why Not Lower Score:**

N/A

**Metareview: Summary, Strengths And Weaknesses:**

This paper proposes a multimodal joint embedding transformer for molecule generation. Specifically, it leverages both text descriptions, molecular string descriptors, and molecular properties to train a join embedding and introduces prompt building on the success of language models. After training such a model, the user can use prompt for a variety of molecule generation tasks, such as property optimization, isomer generation, etc.

Strength of the paper:
1. Prompt-based generation model for molecule is interesting and there is benefit.
2. Experimental results show that MolJET can achieve certain promising results.
3. The model is public.

Weakness of the paper:
1. Reviewers are concerned about the experiments, whether the results are convincing or not. Missing baselines.
2. Zero-shot is unclear.

Overall, this paper still has major limitations and is not ready for publish at the moment.
I would encourage the authors revise according to all the points raised by reviewers.

**Summary Of Ac-Reviewer Meeting:**

N/A